Effects of short-term grazing prohibition on soil physical and chemical properties of meadows in Southwest China

Zhu Guiqing 1
Yuan Chaoxiang 1
Gong Hede gonghede3@163.com 1
Peng Yanling 2
Huang Changjiang 3
Wu Chuansheng wwccss521@163.com 3
Duan Huachao 4
1 School of Geography and Ecotourism, Southwest Forestry University , Kunming , Yunnan , China
2 Technology Department, Southwest Forestry University , Kunming , Yunnan , China
3 Anhui Province Key Laboratory of Environmental Hormone and Reproduction, Anhui Province Key Laboratory of Embryo Development and Reproductive Regulation, Fuyang Normal University , Fuyang , Anhui , China
4 College of Forestry, Southwest Forestry University , Kunming , Yunnan , China
Khan Amanullah
Electronic publication date: 2021 Jun 8
Publication date: 2021
Volume: 9
Electronic Location ID: e11598
Received 2021 Feb 22; Accepted 2021 May 21
Copyright: ©2021 Zhu et al.
Copyright year: 2021
Copyright holder: Zhu et al.
License: This is an open access article distributed under the terms of the Creative Commons Attribution License, which permits unrestricted use, distribution, reproduction and adaptation in any medium and for any purpose provided that it is properly attributed. For attribution, the original author(s), title, publication source (PeerJ) and either DOI or URL of the article must be cited.
License URL: https://creativecommons.org/licenses/by/4.0/

Keywords: No grazing, Plant and soil indexes, Meadow types, NMDS

Funding: National Science and Technology Development Service Project YDCOF20190991 National Natural Science Foundation of China 31600390 Scientific Research Fund Project of the Yunnan Provincial Department of Education 2020Y0392 2020J0409 This work was supported by the National Science and Technology Development Service Project (No. YDCOF20190991), the National Natural Science Foundation of China (No. 31600390) and the Scientific Research Fund Project of the Yunnan Provincial Department of Education (No. 2020Y0392, 2020J0409). The funders had no role in study design, data collection and analysis, decision to publish, or preparation of the manuscript.

==============================
Background

Grassland plays an important role in the ecosystem, but overgrazing harms the grassland system in many places. Grazing prohibition is an effective method to restore grassland ecosystems, and it plays a great role in realizing the sustainable development of grassland systems. Therefore, it is necessary to carry out research on the influence of regional grazing prohibition on the physical and chemical properties of different grassland systems.

Methods

In Potatso National Park, Southwest China, we selected experimental plots in the artificial grazing meadow area to study the effects of grazing prohibition on plant and soil indexes in subalpine meadows and swamp meadows. We investigated the biomass and species diversity of grazing prohibition treatment and grazing treatment plots and sampled and tested the soil index. The variation percentage was used to remove the original heterogeneity and yearly variation, allowing us to compare differences in plant index and soil index values between grazing prohibition and grazing treatments.

Results

Grazing prohibition increased the aboveground biomass, total biomass, total meadow coverage, average height, richness index, Shannon diversity index and evenness index and reduced the belowground biomass and root/shoot ratio in the subalpine meadow and swamp meadow. Additionally, grazing prohibition reduced the pH and soil bulk density and increased the soil total carbon, soil organic carbon, soil total nitrogen, soil hydrolyzable nitrogen, soil total phosphorus and soil available phosphorus in the subalpine meadow and swamp meadow. Nonmetric multidimensional scaling (NMDS) analysis showed that both plant indexes and soil indexes were significantly different between grazing and grazing prohibition treatments and between meadow types. Short-term grazing prohibition had a great impact on improving the fertility of meadow soil in the study area. We suggest that long-term and extensive research should be carried out to promote the restoration and sustainable development of regional grassland systems.

Introduction

Grassland occupies 1/3 of the global land area (Houghton, 1994) and is a major foundation for the development of animal husbandry in many countries; however, grassland resources are facing tremendous pressure with continuous animal husbandry expansion. Previous studies have shown that overgrazing changes surface vegetation and soil nutrient cycling (Smoliak, Dormaar & Johnson, 1972; Chaneton & Lavado, 1996) and can even cause permanent degradation of grassland productivity and ecosystem destruction (Su et al., 2004). Therefore, researchers have conducted a wide range of studies on the impact of grazing on grassland degradation from multiple angles. These include soil nutrients (Jeddi & Chaieb, 2010; Xiong et al., 2014; Ma, Ding & Li, 2016; Li et al., 2016), vegetation (Zhao et al., 2009; Jeddi & Chaieb, 2010; Cheng et al., 2011; Niu et al., 2018) and soil structure (Li et al., 2011b; Jaweed, Saptarshi & Gaikwad, 2012; Mofidi et al., 2012). China has a grassland area of 4.32 × 106 km2, which accounts for 40% of the country’s land area (Tong, Zhao & Wu, 2018), and 90% of grasslands have been degraded to varying degrees. The degradation trend continues to increase (Ren et al., 2007; Cao et al., 2013). Grazing prohibition is considered an effective way to prevent vicious grassland degradation cycles and restore grassland ecosystems and soil fertility (Wu et al., 2014; Bi et al., 2018). Therefore, it is necessary to carry out grazing prohibition work in grassland areas to protect grassland resources and maintain ecological balance.

Studies have found that the plant index is significantly correlated with grazing pressure (Larreguy, Carrera & MB, 2014), and grazing prohibition can significantly increase aboveground and belowground biomass (Cheng et al., 2011; Wu et al., 2014; Xiong et al., 2014; Li et al., 2016), plant coverage, plant richness (Pei, Fu & Wan, 2008; Cheng et al., 2011; Chen & Tang, 2016), total biomass, and average height (Pei, Fu & Wan, 2008).This may be attributed to the grazing prohibition hindering external disturbance activities, protecting the grassland crust structure, and providing opportunities for vegetation growth and reproduction. Moreover, grazing prohibition significantly reduces the root/shoot ratio, Shannon diversity index and evenness (Wang et al., 2014). This is caused by rapid growth of the aboveground parts of the vegetation and the consumption of root nutrient reserves. Grazing prohibition has a protective effect on dominant species, and the constructive species compete for space and nutrients and crowd out other dwarf species. However, studies have also found that grazing prohibition had no effect on belowground biomass (Larreguy, Carrera & MB, 2017) or reduced (Smoliak, Dormaar & Johnson, 1972).

Studies have shown that grazing prohibition significantly reduces pH (Pei, Fu & Wan, 2008; Wu et al., 2010; Wang et al., 2014). This is likely due to grazing prohibition blocking livestock manure and urine input, and dominant species development makes saline-loving vegetation expansion difficult. Meanwhile, grazing prohibition eliminates the soil compaction process caused by livestock trampling and reduces soil bulk density (Hiernaux et al., 1999; Pei, Fu & Wan, 2008; Wu et al., 2010; Wang et al., 2014). Some studies have also found that grazing prohibition can increase, decrease, or not significantly affect soil pH and soil bulk density because of study area differences (Moussa, Rensburg & Kellner, 2009; Lu et al., 2015). Furthermore, grazing prohibition can effectively increase soil organic carbon and soil total nitrogen (Wu et al., 2008; Xiong et al., 2014; Lu et al., 2015; Li et al., 2016), soil total phosphorus and soil available phosphorus (Hiernaux et al., 1999; Li et al., 2011b; Ma, Ding & Li, 2016) because grazing prohibition can largely compensate for soil index losses caused by overgrazing and restore soil ecological indicators in pastoral areas (Raiesi & Riahi, 2014). Studies have found that the effect of grazing prohibits on soil organic carbon (Moussa, Rensburg & Kellner, 2009) and available phosphorus (Lu et al., 2015) without a significant increase or decrease. The same soil index has different responses to grazing prohibition and is related to the terrain, species composition, climate and precipitation of the study area (Wu et al., 2010; Xiong et al., 2014; Zhang et al., 2018).

In summary, the grazing prohibition effects on grassland plant and soil indexes are unclear, and it is necessary to carry out grazing prohibition experiments in more meadow areas to identify the influential mechanism. Meadows, as an important part of grasslands, require strengthened real-time monitoring of degraded areas and carry out grazing prohibition for sustainable development. Subalpine meadow and swamp meadow ecosystems are typical grasslands in Potatso National Park (Wang, Zhong & Yang, 2000); however, nomadic customs have caused degradation in park grasslands. Carrying out this research is an effective means to deeply understand the impact on soil and plant indexes by grazing prohibition in Potatso National Park, and can provide scientific guidance for regional grassland protection and future development.Therefore, we assume that grazing prohibition has a positive effect on the plant and soil indexes of subalpine meadows and swampy meadows. To verify this hypothesis, we conducted grazing and grazing prohibition experiments in the subalpine and swampy meadow pastoral areas of Potatso National Park to determine the effect of grazing prohibition on improving plant and soil indexes.

Materials & Methods

Site description

The study is located in Potatso National Park (27°55′3″N, 99°56′33″E; 3,601 m above sea level) in Shangri-La County, Diqing Tibetan Autonomous Prefecture, northwestern Yunnan, China (Fig. 1A). According to long-term meteorological monitoring data, the annual average temperature is 5.4 °C, the hottest monthly average temperature is 13.2 °C, and the coldest monthly average temperature is −3.8 °C. The average annual precipitation is 619.9 mm, and the summer and autumn precipitation accounts for 80% to 90% of the annual precipitation, respectively (June–September) (Tang & Yang, 2014; He et al., 2019). The soil is mainly dark brown soil, and the bare bedrock on the surface is mostly mica schist with crystalline limestone (Li et al., 2013). The area is rich in animal and plant resources, including 279 species of vertebrates, 67 species of mammals, 171 species of birds, and 13 species of amphibians. The vegetation includes six vegetation types, 11 vegetation subtypes, 34 communities, and more than 2,000 wild seed plants (Wang, Zhong & Yang, 2000). The vegetation types of subalpine meadows and swamp meadows are both alpine meadows (Li et al., 2020). In the subalpine meadow experimental area, Blysmus sinocompressus is the dominant species in the community, and the community species composition includes Blysmus sinocompressus, Carex muliensis, Polygonum viviparum, Potentilla griffithii, Potentilla stenophylla, Stellera chamaejasme, Aletris pauciflora, Gentiana crassula, Veratrilla baillonii and Gentiana wardii. In the swamp meadow experimental area, Blysmus sinocompressus is the dominant species in the community too, and the community species composition includes Blysmus sinocompressus, Carex muliensis, Polygonum viviparum, Potentilla griffithii, Parnassia delavayi, Adonis brevistyla, Plantago depressa, Elsholtzia ciliata, Primula sinopurpurea, Agrimonia pilosa, Prunella vulgaris.

Figure 1 Geographical location and our experiment study site.

Geographical location and our experiment study site. Research location (red dot) (A); grazing prohibition experiment design (B).

Experimental design

With reference to a previous experimental research design, a similar experimental design was adopted in Potatso National Park to carry out the grazing prohibition experiment (Wu et al., 2010; Wang et al., 2014; Lu et al., 2015; Chen & Tang, 2016). In Potatso National Park, typical subalpine meadow and swamp meadow were selected, and six 5 m × 5 m plots were randomly selected in both subalpine meadow and swamp meadow types. The distance between plots in the same meadow types was 2 m (Fig. 1B). In the subalpine meadow, three plots surrounded by barbed wire fence were used for the grazing prohibition treatment, and three grazing treatment plots were randomly selected from free grazing areas; we used the same method to set up the experimental plots in the swamp meadow as in the subalpine meadow. In this experiment, the grazing plots were perennial grazing plots with a grazing history of more than 100 years, and the grazing animals were yaks and horses. The grazing prohibition plots were short-term complete grazing prohibition plots, and the grazing prohibition period was 1 year (September 2019–September 2020). In this study, the grazing pressure was the same for the subalpine meadow and swamp meadow. Every morning, livestock departed from the swamp meadows to the subalpine meadows, passed through the woods, and returned on the same road in the evening, passing through the woods, subalpine meadows and swamp meadows in turn.

After the grazing prohibition treatment (the growth of the aboveground biomass was stable), three 1 m × 1 m square sampling areas were selected in each of the 12 sample plots, and the aboveground plants were trimmed close to the ground and dried to a constant weight to estimate the aboveground biomass. Three 50 cm × 50 cm × 20 cm soil samples were taken from each experimental plot, three repeated samples from each plot were dried to constant weight, and the average value was taken to estimate the belowground biomass in every plot. The sampling point distance was greater than 1 m, and foreign matter (soil and stones) was washed and removed. An earth-boring drill (diameter 3.8 cm, volume 100 cm3) was used sample the following soil layers: 0–10 cm, 10–20 cm, 20–30 cm, 30–40 cm, and 40–60 cm (note: at 60 cm, the drill reached the rock layer and further sampling could not be performed). We sampled 3 random sites in an area of approximately 1 m × 1 m , mixed the same soil layer samples taken from 3 sampling sites, sealed them and sent them to the laboratory for air drying and root removal treatment. Then, the samples were analyzed for pH, soil bulk density, soil total carbon, soil organic carbon, soil total nitrogen, soil hydrolyzable nitrogen, soil total phosphorus and soil available phosphorus. In each treatment plot, three 1 m × 1 m square areas were delineated along the diagonal from the upper left corner, the middle, and the lower right corner to investigate the plant index, and the collected specimens were brought back to the laboratory for identification.

Data collection and calculations

The calculation formulas for the meadow community richness index, Shannon diversity index and evenness index in the state of grazing prohibition and free grazing in subalpine meadows and swamp meadows are as follows Wu et al. (2009) and Wang et al. (2014):

Richness index (R): (1) R=S

Shannon diversity index (H): (2) H=−∑i=1SPilnPi

Evenness index (E): (3) E=HlnS

where S is the total number of species in the meadow community per unit area and Pi is the proportion of the species in the total species.

Field experiment plots are prone to differences. To remove the original heterogeneity and yearly variation, we calculated the variation percentage (V, %) of the soil indexes and plant indexes with the following formula: (4) Vi=FinalInitial−1×100%

where i refers to the soil indexes and plant indexes. To calculate the grazing prohibition effects (GPE), the formula was as follows: (5) GPEi=Vigp−Vig

where gp refers to grazing prohibition and g refers to grazing.

Then, the Vi between grazing and grazing prohibition in each meadow type was tested by the t test or Wilcox test (when data were not normally distributed). Nonmetric multidimensional scaling (NMDS) was used to simplify the meadow types and grazing treatment samples to a low-dimensional space for positioning, analysis and classification while retaining the original relationship between the objects. All data are expressed as the mean ± standard deviation (Mean ± SD), and all statistical tests and figure drawing were performed using R software and Rstudio (R version 4.0.3 for Windows, using packages readxl, ggplot2, ggpubr and vegan).

Results

Effects of grazing prohibition on plant indexes

In the subalpine meadow, grazing prohibition significantly increased aboveground biomass (p < 0.001; Fig. 2A), total meadow coverage (p < 0.01; Fig. 2E) and average height (p < 0.001; Fig. 2F) by 324.96, 25.35 and 225.56%, respectively, and the total biomass, richness index, Shannon diversity index and evenness index increased by 17.51, 16.67, 20.62 and 11.05%, respectively, but without statistical significance (Figs. 2C, 2G, 2H & 2I). However, grazing prohibition significantly reduced the root/shoot ratio by 101.65% (p < 0.001; Fig. 2D), while the reduction in belowground biomass, at 22.07%, was not significant (Fig. 2B).

Figure 2 Effect on plant indexes.

Effect on plant indexes. Above-ground biomass (A), below-ground biomass (B), total biomass (C), root-shoot ratio (D), total meadow coverage (E), average height (F), Richness Index (G), Shannon Diversity Index (H) and Evenness Index (I) in subalpine meadow and swamp meadows between grazing and grazing prohibition treatment. significant differences between grazing and grazing prohibition treatment are indicated by symbols: ***p < 0.001, **p < 0.01, *p < 0.05; and no symbol, no significant difference.

In the swamp meadow, grazing prohibition significantly increased the aboveground biomass (p < 0.001; Fig. 2A), total biomass (p < 0.01; Fig. 2C), total meadow coverage, average height, Shannon diversity index and evenness index (p < 0.05; Figs. 2E, 2F, 2H & 2I), and these values increased by 283.92, 79.99, 4.27, 132.79, 72.89 and 54.15%, respectively. The richness index increased by 26.98% but without statistical significance (Fig. 2G). Grazing prohibition significantly reduced the root/shoot ratio by 54.20% (p < 0.001; Fig. 2D), but the reduction in belowground biomass of 11.54% was not significant (Fig. 2B).

NMDS analysis showed that plant indexes between grazing and grazing prohibition were significantly different (stress =0.097), and those between meadow types were not different (Fig. 3A).

Figure 3 NMDS analyzes the community structure of plant and soil samples.

NMDS analyzes the community structure of plant and soil samples. Effects on plant community index (A) and soil index (B) in subalpine meadow and swamp meadow between grazing and grazing prohibition treatment. The shape represents meadow types, the color represents different treatments, the soil depth is represented by the size, and the circle represents the 95% confidence interval of the centroid position of each group. The NMDS analysis is based on Bray-Curtis similarity, and the stress value is shown in the lower right corner.

Effect of grazing prohibition on soil indexes

In the subalpine meadow, grazing prohibition reduced the pH by 0.70 to 1.53% (Fig. 4A) and the soil bulk density by 3.91 to 15.95% (Fig. 4B). Grazing prohibition increased the soil total carbon, soil organic carbon, soil total nitrogen, soil hydrolyzable nitrogen, soil total phosphorus and soil available phosphorus (Figs. 5A, 5B, 6A, 6B, 7A & 7B), and these values increased by 2.02 to 28.93%, 2.10 to 16.35%, 1.09 to 11.66%, 3.35 to 13.03%, 0.64 to 5.36% and 0.17 to 3.74%, respectively. In addition, the increases in soil total carbon and soil organic carbon were significant in the 0–20 cm and 10–20 cm layers (p < 0.05; Figs. 5A & 5B), respectively. The other parameters were not significantly different between grazing and grazing prohibition treatments.

Figure 4 Changes of pH and soil bulk density with soil depth in subalpine meadow and swamp meadow between grazing and grazing prohibition treatment.

Changes of pH (A) and soil bulk density (B) with soil depth in subalpine meadow and swamp meadow between grazing and grazing prohibition treatment. significant differences between grazing and grazing prohibition treatment are indicated by symbols: ***p < 0.001, **p < 0.01, *p < 0.05; and no symbol, no significant difference.

Figure 5 Changes of soil total carbon and soil organic carbon with soil depth in subalpine meadow and swamp meadow between grazing and grazing prohibition treatment.

Changes of soil total carbon (A) and soil organic carbon (B) with soil depth in subalpine meadow and swamp meadow between grazing and grazing prohibition treatment. significant differences between grazing and grazing prohibition treatment are indicated by symbols: ***p < 0.001, **p < 0.01, *p < 0.05; and no symbol, no significant difference.

Figure 6 Changes of soil total nitrogen and soil hydrolyzable nitrogen with soil depth in subalpine meadow and swamp meadow between grazing and grazing prohibition treatment.

Changes of soil total nitrogen (A) and soil hydrolyzable nitrogen (B) with soil depth in subalpine meadow and swamp meadow between grazing and grazing prohibition treatment. significant differences between grazing and grazing prohibition treatment are indicated by symbols: ***p < 0.001, **p < 0.01, *p < 0.05; and no symbol, no significant difference.

Figure 7 Changes of soil total phosphorus and soil available phosphorus with soil depth in subalpine meadow and swamp meadow between grazing and grazing prohibition treatment.

Changes of soil total phosphorus (A) and soil available phosphorus (B) with soil depth in subalpine meadow and swamp meadow between grazing and grazing prohibition treatment. significant differences between grazing and grazing prohibition treatment are indicated by symbols: ***p < 0.001, **p < 0.01, *p < 0.05; and no symbol, no significant difference.

In the swamp meadow, grazing prohibition reduced the pH and soil bulk density by 1.06 to 2.70% and 2.22 to 6.87% (Figs. 4A & 4B), respectively. Grazing prohibition increased the soil total carbon, soil organic carbon, soil hydrolyzable nitrogen, soil total phosphorus, soil available phosphorus and 0–30 cm soil total nitrogen (Figs. 5A, 5B, 6A, 6B, 7A & 7B), and these values increased by 25.91 to 117.23%, 9.02 to 48.30%, 19.61 to 71.80%, 8.30 to 40.50%, 17.51 to 52.87% and 8.26 to 30.85%, respectively. The increases were significant for soil total carbon (p < 0.05 and 0.01), soil hydrolyzable nitrogen and soil available phosphorus at 0–60 cm (p < 0.05, 0.01 and 0.001); soil organic carbon at 10–40 cm (p < 0.05); and soil total phosphorus at 20–40 cm (p < 0.01 and 0.001).

NMDS analysis showed that the soil index values were significantly different between grazing and grazing prohibition sites (stress =0.178) but that the soil index values were not different between meadow types and among soil depths (Fig. 3B).

Discussion

Grazing prohibition is an effective means to improve grassland vegetation parameters (Pei, Fu & Wan, 2008; Cheng et al., 2011; Chen & Tang, 2016; Li et al., 2016). In our study, grazing prohibition treatments increased the aboveground biomass, total biomass, total meadow coverage, average height, richness index, Shannon diversity index and evenness index in both the subalpine meadow and swamp meadow (Figs. 2A, 2C, 2E, 2F, 2G, 2H & 2I), consistent with the results of other studies (Jeddi & Chaieb, 2010; Cheng et al., 2011; Chen & Tang, 2016; Oñatibia, Boyero & Aguiar, 2018). This may be because the grazing prohibition treatment protected vegetation from gnawing and trampling by livestock, there more grass when livestock disturbance was reduced (Wu et al., 2010). Our grazing prohibition experiment reduced the belowground biomass (Fig. 2B), which was different from other research results (Wu et al., 2010; Wu et al., 2014; Li et al., 2016). Previous researchers suggested that the belowground biomass increases may be due to the severe degradation of grassland caused by overgrazing in the study area and that following restoration of grassland vegetation in the grazing prohibition treatment, the vegetation root system became more developed than that prior to the grazing prohibition (Wu et al., 2014). Our results are consistent with other study results (Smoliak, Dormaar & Johnson, 1972): the decrease may be due to the reduction of external interference and the transfer of soil-derived nutrients from vegetation roots to the aboveground part for growth. In contrast, in the grazing area, vegetation is prone to exhibiting self-protection behavior when threatened by gnawing and trampling and uses nutrients for root growth to guarantee survival (Smoliak, Dormaar & Johnson, 1972). The grazing prohibition treatment significantly increased aboveground biomass and reduced belowground biomass, which in turn significantly reduced the root/shoot ratio (Fig. 2D). The increase or decrease effect of the plant index after grazing prohibition differed between the meadow types (Fig. 3A), and this difference was mainly driven by the different meadow types and species compositions (Milchunas, Lauenroth & Burke, 1998; Qiu et al., 2013; Wang et al., 2014).

Grazing prohibition reduced pH and soil bulk density (Figs. 4A & 4B), which was consistent with other research results (Smoliak, Dormaar & Johnson, 1972; Hiernaux et al., 1999; Wang et al., 2014; Ma, Ding & Li, 2016). This result may be because grazing prohibition treatments isolate livestock activities, hinder the input of excrement and the impact of livestock trampling. The dominant species supplants saline-loving vegetation and effectively improves the soil structure, thereby reducing pH and soil bulk density (Pei, Fu & Wan, 2008; Wang et al., 2014). Grazing prohibition is an effective means to increase soil total carbon, soil organic carbon (Wu et al., 2008; Piñeiro et al., 2009; Wu et al., 2010; Rui et al., 2011), soil total nitrogen (Raiesi & Riahi, 2014; Xiong et al., 2014; Ma, Ding & Li, 2016), soil total phosphorus and soil available phosphorus (Hiernaux et al., 1999; Li et al., 2011b; Ma, Ding & Li, 2016). In our study, the grazing prohibition treatment increased soil total carbon, soil organic carbon, soil total nitrogen, hydrolyzable nitrogen, soil total phosphorus and soil available phosphorus in both the subalpine meadow and swamp meadow (Figs. 5A, 5B, 6A, 6B, 7A & 7B), which was consistent with most research findings (Pei, Fu & Wan, 2008; Lu et al., 2015; Li et al., 2016; Ma, Ding & Li, 2016). Some values of these soil indexes show significant differences between the grazing prohibition and grazing treatments, such as 0–20 cm soil total carbon (Fig. 5A; p < 0.05) and 10–20 cm soil organic carbon (Fig. 5B; p < 0.05) in the subalpine meadow and 0–60 cm total soil carbon (Fig. 5A; p < 0.05), 10–40 cm soil organic carbon (Fig. 5B; p < 0.05), 0–60 cm hydrolyzable nitrogen (Fig. 6B; p < 0.05), 20–40 cm soil total phosphorus (Fig. 7A; p < 0.01) and 0–60 cm soil available phosphorus (Fig. 7B; p < 0.05) in the swamp meadow. Some of these differences are significant at the p < 0.001 level. These soil index changes may be due to the grazing prohibition treatment compensating for the loss caused by grazing (Fuhlendorf et al., 2002; Raiesi & Riahi, 2014; Larreguy, Carrera & MB, 2014). The grazing prohibition treatment reduced the soil total nitrogen in the swamp meadow at 30–60 cm and may be related to the higher soil gravel content of the soil layer (30–60 cm) in the swamp meadow.

Through comparison, grazing time and grazing intensity have a significant impact on plant diversity and community structure (Niu et al., 2018), mainly due to the removal of plants by animal gnawing and trampling (Wen et al., 2013; Wang et al., 2014). Previous studies have shown that light and moderate grazing can effectively restore the richness index and Shannon diversity index in degraded grasslands, while overgrazing reduces these indexes (Zhao et al., 2009). Light grazing increases the plant coverage (Li et al., 2011a; Larreguy, Carrera & MB, 2014), richness index (Niu et al., 2018; Oñatibia, Boyero & Aguiar, 2018), total biomass (Niu et al., 2017), aboveground biomass and plant height (Li et al., 2011a). This likely occurs because light grazing represents a reduction in livestock gnawing and trampling, reduces livestock destruction of the grass crust, and provides dwarf and more disturbed species greater chances of survival. Simultaneously, light grazing also reduces belowground biomass and the root/shoot ratio (Li et al., 2011a), which can be attributed to the adaptive response of grassland vegetation to changes in grazing pressure. Long-term grazing prohibition can effectively improve the characteristics of grassland communities (Cheng et al., 2011). Study results from the Alxa Desert indicated that the plant coverage, grass height, and aboveground biomass increased by 47.55, 109.77 and 58.51%, respectively, at six-year prohibition sites. The increases at the two-year prohibition sites were 21.99, 89.19 and 9.57%, respectively (Pei, Fu & Wan, 2008), and these increases were higher than those at the grazing sites. A study in southern Tunisia produced analogous results: a twelve-year grazing prohibition treatment allowed the plant coverage to increase by 2 times relative to grazed sites, with a percentage increase of 71.3%, and the aboveground biomass was 3 and 3.4 times higher in six-year and twelve-year prohibition sites, respectively, than that at grazed sites (Jeddi & Chaieb, 2010). Obviously, differences in the study area, grazing prohibition history, and grazing intensity result in different effects.

Grazing pressure has a large effect on soil indexes, and livestock damage and excrement can easily lead to soil salinization, which in turn increases the spread of saline-loving plant communities and leads to increases in soil pH (Wu et al., 2010; Wang et al., 2014). Livestock trampling causes soil compaction (Hiernaux et al., 1999), which greatly increases soil bulk density (Mofidi et al., 2012). Long-term (nine-year) grazing prohibition significantly altered soil properties, and the pH and soil bulk density were reduced by 6.53 and 30.23% when compared with the grazing treatment (Wu et al., 2010). This value was higher than our results (Figs. 4A & 4B), likely due to the effects of grazing prohibition history. The Alxa Desert steppe results illustrate that the soil buck density decreased by 4.08 to 5.33% and 3.27 to 5.52% at the six-year and two-year prohibition sites, respectively, and the pH decreased by 2.45 to 3.21% and 1.35 to 2.24%, respectively (Pei, Fu & Wan, 2008). There were differences in the reduction, and our results may be due to differences in the meadow types, grazing prohibition histories, grazing intensities, species compositions and initial index values. In this study, the mean increase in soil total carbon (11.73%) in the subalpine meadow was less than the value of 17% observed in the Zagros Mountains in central Iran (Raiesi & Riahi, 2014) (Fig. 5A), and the mean increase in soil organic carbon (6.32%) was also lower than the value of 13.9% observed in the Hulunbuir grassland (Wu et al., 2014) and the value of 22% observed in the Alxa Desert grasslands after six years of grazing prohibition (Pei, Fu & Wan, 2008) (Fig. 5B). However, the mean increases in the swamp meadow’s soil total carbon (83.29%) and soil organic carbon (25.03%) were obviously higher than those observed in these other studies (Figs. 5A & 5B). Except for individual soil depths, the grazing prohibition experiment effect on the mean increases in soil total nitrogen in the subalpine meadow (6.72%) and swamp meadow (7.56%) was weaker than the values observed by Pei, Fu & Wan (2008) , i.e.,14%, Raiesi & Riahi (2014), i.e., 19%, and Ma, Ding & Li (2016), i.e., 27.4% (Fig. 6A). Moreover, different conclusions were reached regarding the impact of grazing prohibition on soil total phosphorus and soil available phosphorus. Li et al. (2011a) concluded that light grazing can reduce soil total phosphorus and soil available phosphorus, but most research results have shown that grazing prohibition increases soil total phosphorus and soil available phosphorus (Hiernaux et al., 1999; Li et al., 2011b; Ma, Ding & Li, 2016). Studies have also found that grazing prohibition reduces soil total phosphorus (Lu et al., 2015) or has no effect (Li et al., 2018) and has no effect on soil available phosphorus (Lu et al., 2015). In our study, grazing prohibition increased the soil total phosphorus and soil available phosphorus in the subalpine meadows and swamp meadows (Figs. 7A & 7B). Through our experiments, we also found that the effects of grazing prohibition on soil indexes were different among different meadow types (Fig. 3B). This result also showed that the grazing prohibition area and time length were different, which led to the different effects of the grazing prohibition treatment on the soil indexes.

Our research results differed between the two meadow types (Figs. 3A & 3B), and there was also a certain difference between the research results from different regions. This difference may be affected by the terrain (Zhang et al., 2018), soil structure (Jaweed, Saptarshi & Gaikwad, 2012), surface species (Wu et al., 2010; Wang et al., 2014) and natural precipitation conditions (Xiong et al., 2014). Our results suggested that short-term grazing prohibition played a positive role in the plant and soil indexes in the free grazing area of Potatso National Park. The grazing prohibition effects were weaker than those of long-term treatments, and the results also showed that we need to carry out long-term experimental research in the region. In short, overgrazing is unconducive to the physical and chemical properties of soil and the sustainable development of surface vegetation, and grazing prohibition is beneficial for improving plant and soil indexes (Bi et al., 2018). Scientific and effective management have the potential reverse this degradation. While improving soil fertility, grazing prohibition also protects regional species diversity. In addition, we suggest carrying out rotation grazing and nitrogen deposition experiments in degraded areas to better utilize the value of limited grassland resources.

Conclusions

This study’s main purpose was to evaluate the effectiveness of a grazing prohibition treatment on restoring degraded areas in the free pasture area of the subalpine meadow and marsh swamp ecosystems in Potatso National Park, China. We used plant and soil indicators as the main research objects. According to the experimental results, we found that the grazing prohibition treatment was effective in restoring the surface plant index, and it increased the aboveground biomass, total biomass, total meadow coverage, average height, richness index, Shannon diversity index and evenness index in both subalpine meadows and swamp meadows. In contrast, it reduced the belowground biomass and root/shoot ratio. There were differences between meadow types, but the grazing prohibition effect was more important. Meanwhile, the grazing prohibition treatment improved the soil index, reduced the pH and soil bulk density, and increased the soil total carbon, soil organic carbon, soil total nitrogen, soil hydrolyzable nitrogen, soil total phosphorus and soil available phosphorus in the subalpine meadow and swamp meadow. NMDS analysis showed that different meadow types and soil depths had an impact on the soil index, but the impact was less than that of the grazing prohibition treatment.

Supplemental Information

Supplemental Information 1 Raw data on grazing prohibition effect

Data analysis shows that grazing prohibition can improve plant and soil indexes of subalpine meadows and swamp meadows, and compensate for the damage of plant and soil indexes caused by overgrazing.

Click here for additional data file.

This work is a research contribution from Potatso National Park for Meadows Ecosystem Studies.

Additional Information and Declarations

Competing Interests

Author Contributions

Data Availability

The authors declare there are no competing interests.

Guiqing Zhu and Chaoxiang Yuan conceived and designed the experiments, performed the experiments, analyzed the data, prepared figures and/or tables, authored or reviewed drafts of the paper, and approved the final draft.

Hede Gong conceived and designed the experiments, prepared figures and/or tables, authored or reviewed drafts of the paper, provide paper layout fee, and approved the final draft.

Yanling Peng, Changjiang Huang and Huachao Duan analyzed the data, prepared figures and/or tables, and approved the final draft.

Chuansheng Wu analyzed the data, prepared figures and/or tables, authored or reviewed drafts of the paper, and approved the final draft.

The following information was supplied regarding data availability:

The raw measurements are available in the Supplementary File.

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
