# Peer review of "Effects of short-term grazing prohibition on soil physical and chemical properties of meadows in Southwest China"

_PeerJ, doi:10.7717/peerj.11598_

## Round 0.1 · original submission · Major Revisions

Dear authors

Revise your article according to the comments and suggestions of reviewers. Show the changes in the manuscript in red. Also send separate file having your replies to each comment of each reviewer.

Include new and updated literature.

Improve the English in the whole manuscript.

Critical discussion is needed in relation to updated literature.

Explain your experimental design in detail. Treatments/replications/analysis of variance/degree of freedom, and design used along with a reference.

Reviewer 1 ·

Basic reporting

Clear and unambiguous, professional English used throughout.
Most part was clear and professional.

Literature references, sufficient field background/context provided.
Making a stronger explanation for the aim of this study could be better.

Professional article structure, figures, tables. Raw data shared.
Please check the resolution of all figure 1 to 5 are according with the journal requirements (often at least 220 dpi or 300 dpi). Many figures were blurry.

Self-contained with relevant results to hypotheses.
The submission should be ‘self-contained,’ should represent an appropriate ‘unit of publication’, and should include all results relevant to the hypothesis.
Yes.

Coherent bodies of work should not be inappropriately subdivided merely to increase publication count.
Some part could be more concise, please see Minor issues 2.

Experimental design

This study land on aims and scope of PeerJ. Effect of grazing prohibition on soil prosperity and vegetation has been published a lot. However, this study could be another useful example in this field. Methods were described with many details and information. However, there could be an important detail about grazing prohibition period that should be definite. Please see the Major issues 1. Moreover,it is necessary to give a map or sketch of study area, and graph of experimental design with plots.

Validity of the findings

According to this study, the findings were statistically. However, some details could be necessary to confirm that the result was robust. Please see the Major issues 2.

Additional comments

This study was carried out in a national part of China, which located in a typical subalpine meadows and swamp meadows area. Researchers study the effect of grazing and grazing prohibition on soil prosperity and vegetation, which is a valuable experiment in the field of nature conservation.

Major issues.
1.According to previous studies, effect of grazing prohibition on soil and vegetation were different, as you mentioned in Discussion session. One of the most vital factors could be how long grazing prohibition was. Hence, it is recommended that describe more details on grazing and grazing prohibition history. Moreover, compared with previous studies with different grazing and prohibition history could be important and make the ms much stronger.

2.Taking grazing treatment with Before as an example. In the method session, both subalpine meadow and swamp meadow has three plots as replicate. And three 1X1m sampling areas were selected in each plot for soil prosperity samples, that is to say, three samples as replicate in each plot for 0-10cm of soil layer were collected. Hence, there were 9 samples as replicate in three plots for 0-10cm of soil layer. Similar with other soil layer samples from 20-60cm However, there only three raw data for 0-10cm soil layer. Do you mean that the raw data of each 0-10cm soil layer showed in the EXCEL file was mean value of three 1X1m sampling areas? Could you please give a specific and scientific describe in Methods session? Because this is vital to make your ms stronger.

Minor issues
1.I think it is a good manuscript, but could be substantially improved after revision with some additional editing by a native English speaker.

2.L227 to L235 can be described more concise.

pecific comments
L146, “in both subalpine” other than “in subalpine” could be less ambiguity. According to the ms, 6 plots in subalpine and 6 plots in swamp meadow. There were 12 plots in total.

L154 Does “Fifty cm × 50 cm × 20 cm” means “50 cm × 50 cm × 20 cm”?

L144 Could you please give some references that used the same or similar methods as yours?

L176, L178, L180, L187 and L190 The format of equations could not be image. Please check the policy of PeerJ.

L197 SD or SE. should be given, besides mean. Mean ± SD or Mean ± SE.

L204 All “P <” should be “P <”.

L207 All “P >” can be deleted for concise.

L208 and L209 “However” and “but” are not usually used together in the same sectence.

L226 In this session, could you please give some reasons that why P >0.05 were showed?

L278 According to Figs. 2, both pH and soil bulk density had no significant decrease after grazing prohibition, compared to grazing. However, changes of pH showed in Ding & Li (Ding & Li, 2016) that you mentioned in your ms increased, which was not consistent with your result. Hence, please check your references if they were available in your ms. Furthermore, the discussion part of pH and soil bulk density should be revised seriously.

L290 According to Figs. 3, Figs. 4 and Figs.5, many TC, SOC, N, TP, soil available phosphorus, soil hydrolysable nitrogen had significant different between grazing and grazing prohibition. P value should be showed in your ms.

L310 In this paragraph, number of figures should be added to make a better readability of your ms.

Reviewer 2 ·

Basic reporting

The manuscript is well prepared, with clear English, literature references, and results presentation. However, I did not see any raw data associated with this MS.

Experimental design

experimental design is quite well. But there is a lack of grazing intensity gradient issue, please see my following comments.

Validity of the findings

Findings were clear, but too descriptive. See my following comments for your consideration.

Additional comments

Zhu et al. conducted a field experiment to assess how grazing prohibition may affect the physical and chemical properties of two types of meadows in southwest china, and they found that short-term grazing prohibition significantly improved soil fertility of the investigated areas. Overall, the topic of this study is quit interesting, as the results of such studies would help us to conserve the degraded grassland due to extensive grazing. This manuscript would be a good contribute to PeerJ, but I have several comments that should be addressed.

1. In general, this manuscript is quite descriptive, the underlying mechanisms that explain the observed phenomenon were not well addressed. For example, plant biomass and diversity were measured, but what’s their relationships with soil properties? Which plant trait may explain the changes in soil properties? Please link your findings and try to found a mechanism;
2. In the introduction section, the effects of light grazing were well discussed, but I did not find any more about this in the following section. Actually, light grazing may be a stimulating factor for grassland productivity. But I did not see a grazing intensity gradient in the experimental design; If you have such data, please add. Or discuss this issue in your manuscript.
3. Also, in the introduction section, please discuss more about the potential mechanism for the positive effects of grazing prohibition on soil properties, not just list the findings of previous studies.
4. The quality of all the figures were quite low, please use high quality figures. For example, insert emf figures in the word would be very clear.

Reviewer 3 ·

Basic reporting

no comment

Experimental design

the material & method part lacks the details of the experimental design and the background of the experimental site.

Validity of the findings

in the result part, the data which showed no significance means no difference, so the results should be re-described.

Additional comments

Comment 1: The significance and importance of this study are not well elaborated in the introduction part.
Comment 2: in line 142, fifty should be 50cm. And how many 50cm*50cm*20cm were sampled in each plot? Please give the detail. And what is the distance between the plots?
Comment 3: the material & method part lacks the details of the experimental design and the sampling. The grazing animal, the grazing intensity, and the grazing system such as seasonal grazing or perennial grazing, all of those should be given in detail in the grazing treatment. And in the grazing prohibition treatment, the detail such as prohibition time, completely grazing prohibition or periodical grazing prohibition should be listed.
Comment 4: in site description part, the vegetation types, dominant species and the species composition in subalpine meadow and swamp meadow should be given respectively.
Comment 5: in the result part, the data which showed no significance means no difference, so the results should be re-described.
Comment 6: The discussion is problematic in that it is poorly organized and lack scientific rigor. The discussion did not take into account the main distinctive results of this study. A large number of other people's research results were enumerated, but it does not have very good explanation and cohesion to the results of this study.
Comment 7: I strongly recommend collaborating with a native English speaker to revise the article to avoid the use of a large number of unnecessary words and sentences which make the article tedious and verbose.

---

## Round 0.2 · accepted · Accept

The manuscript is revised and improved.

Reviewer 2 ·

Basic reporting

I can see that the authors have tried to revise the MS according to my comments, which were well addressed or answered. I think the current version of this MS is worth to be published.

Experimental design

Much clearer now.

Validity of the findings

Good.

Additional comments

I am happy that the authors have well addressed all my comments, the MS is much clearer and worth to be published.